# Prevalence of mental health conditions in post-conflict Kasai Province, Democratic Republic of the Congo: A repeated, cross-sectional study

**Maaike L. Seekles**[1]*, **Jacob K. Kadima**[2], **Pierre O. L. Omumbu**[2], **Junior K. Kukola**[2], **Joy J. Kim**[3], **Christian B. Bulambo**[2], **Raphael M. Mulamba**[2], **Motto Nganda**[1], **Laura Dean**[1]

1 Department of International Public Health, Liverpool School of Tropical Medicine, Liverpool, United Kingdom, 2 The Leprosy Mission DRC, Kinshasa, Democratic Republic of the Congo, 3 Effect:Hope, Markham, Ontario, Canada

* maaike.seekles@lstmed.ac.uk

## Abstract

Globally, one in five people in post-conflict areas are estimated to be living with a mental health condition. As a key public health issue, these conditions negatively affect individuals, communities, and societies to function after a conflict. Documenting the prevalence of mental health conditions amongst these populations is crucial to prioritise and guide future mental health interventions. This study was the first to use a repeated cross-sectional design and sex-disaggregated analysis, with the aim of estimating the prevalence of depression (PHQ-9) and anxiety (GAD-7) in a post-conflict population of the Kasai Province, Democratic Republic of the Congo. Several domains of Quality of life (WHO-QoL-BREF) were also assessed to gain insight into the relationship between bio-psychosocial stressors and mental health status. Using random cluster sampling, data were collected in two waves from 385 participants, with a one-year interval. The pooled prevalence across both waves was 34.3% for major depression disorder and 26.5% for generalised anxiety disorder. Multivariable linear regression analysis showed that depression and anxiety were both predicted by being female, being of older age, and by experiencing lower physical quality of life, but not by the passing of time. For both mental health outcomes, environmental quality of life served as a significant predictor for women, but not for men. In conclusion, these results suggest that a lack of mental health services and continued exposure to daily stressors are linked to a sustained high prevalence of mental health conditions in our study population. There is a significant need for the development of mental health services in the region. These services should go beyond biomedical interventions and include multi-sectoral approaches that consider the social determinants of (mental) health.

## Introduction

A quarter of the world's population is estimated to live in conflict-affected areas [1]. In sub-Saharan Africa, conflict persists within and between many countries; the region is home to five of the ten least peaceful countries in the world [2]. One of these countries is the

**Data availability statement:** The dataset and analysis codes have been uploaded as Supporting Information files. Please note that the data related to participant age has been removed to ensure participant anonymity. The full dataset is available from the authors on reasonable request.

**Funding:** This work was funded by Effect:Hope (https://effecthope.org/). This funding partially covered the salaries of all authors. An employee of the funding body (JK) was part of the project team and contributed to the study design, data collection and manuscript preparation.

**Competing interests:** The authors have declared that no competing interests exist.

Democratic Republic of the Congo (DRC), which has seen protracted violence and ethnic clashes for several decades.

Conflict-affected populations are vulnerable to developing mental health conditions at higher rates than the general population. The impact of conflict on the mental health of individuals, families and communities is complex and multifaceted. Direct exposure to conflict-related traumatic events, chronic stress, grief, and feelings of hopelessness can serve as a catalyst for the emergence of a multitude of new mental health conditions, or the exacerbation of already existing underlying ones. Insecurity, violence and displacement can lead to increased poverty and the breakdown of support networks and basic services such as healthcare, education, housing, water and sanitation. This exposes those affected to increased levels of daily stressors that may evolve into mental health conditions, particularly where mental health support services are absent and/or have limited capacity [3–5]. In the absence of adequate support, individuals may turn to substance abuse as a way to cope, which further disrupts recovery.

A 2019 review and meta-analysis to update World Health Organization (WHO) estimates found that more than one in five people (22%) in post-conflict settings has a mental health condition, such as post-traumatic stress disorder (PTSD), depression, or anxiety [6]. More specifically, several studies in East African post-conflict settings found prevalence estimates ranging from about 25% to 50% for these conditions [7–12]. Mental health conditions are amongst the leading causes of disability worldwide [13] and the mental health effects of living through conflict can be long-lasting and may be transmitted intergenerationally. They can directly affect the ability of individuals, communities, and societies to function after conflict, and may influence respondent attitudes to post-conflict reconciliation [14]. Therefore, recent years have seen increased recognition of the importance of addressing mental health and psychosocial needs in effective and sustainable post-conflict reconstruction [5].

In the Kasai Province of the DRC, a violent conflict took place between 2016 and 2019, initially sparked by a dispute between customary chiefs and the government. Intense fighting erupted and existing inter-community and inter-ethnic tensions became part of a wider conflict that saw child soldiers fighting alongside militias, armed groups, and security forces. At its peak, the conflict led to thousands of deaths, over a million people displaced, and a spike in rape and gender-based violence toward women. Although it has largely subsided, the conflict has exacerbated risk factors for poor mental health in the area: it left pockets of instability and widespread destruction and destitution, with schools and health centres destroyed. It has worsened pre-existing food insecurity and livelihood crises. In addition, there is said to be a profound gender crisis in most parts of Kasai [15,16].

Generally, documenting the prevalence and burden of mental health problems in conflict-affected and post-conflict settings has been identified as a priority in global mental health research [17]. There have been calls for mental health and/or psychosocial programmes in the aftermath of the Kasai conflict [15]. Yet, there are currently no studies that provide data on the mental health of people living in the region to inform future interventions or programming. In response, this paper quantitatively assesses depression and anxiety in the region's population. It uses a sex-disaggregated analysis, in recognition of the differing impacts of conflict on men and women. Depression and anxiety are known to be transient conditions, and it is usually anticipated that mental health challenges may be heightened in an immediate post-conflict period. Thus, we applied a repeated design to gain further understanding into how mental health outcomes might change over time. Additionally, to gain further insight into stressors affecting this population we explored Quality of Life (QoL) domains and their relationship with mental health.

## Methodology

### Ethics statement

Ethical approval for this study was obtained from the Congolese National Health Ethics Committee (reference number: 269/CNES/BN/PMMF/2021) and the Liverpool School of Tropical Medicine (reference number: 21-053). Written informed consent was obtained, via signature or fingerprint.

### Study design and procedures

We used a repeated, cross-sectional study design, with randomised cluster sampling. The survey took place in April 2022 (Wave 1) and April 2023 (Wave 2) in Tshikapa, the capital of Kasai Province, which was heavily affected by the conflict. Within Tshikapa health zone, two health areas (Ngombe, population ~19,300 and Tshisele, population ~13,600) were randomly selected. Using a direct estimation method for the number of clusters [18], and available population sampling frames in both health areas, we determined the number of clusters to be 23. Using an expected prevalence of depression of 11% [6], a level of acceptable error of 5% and a 5% Type I error rate, we obtained a required sample size of 151 [19]. We aimed to survey the same number of participants per sex in each cluster, resulting in a target sample of 4 male and 4 female participants per cluster (total target sample 184). Any person over the age of 18 years was eligible to participate. Data was collected at two time points in April 2022 and April 2023. Data collectors received training in basic psychological support and psychological first aid, delivered by a Congolese psychologist. Due to low literacy rates, data collectors described information sheets verbally and explained to participants that a psychologist was available if they wished to be referred. The psychologist delivered one community-based mental health awareness session after the data collection was completed. To the best of our knowledge, no post-conflict reconstruction interventions (including psychosocial interventions) have been delivered in the area in the time period between both waves. The survey was administered verbally and directly entered in a tablet using Redcap software.

### Measurements

In addition to questions that captured sociodemographic data (sex, age, employment situation), the survey consisted of three questionnaires:

The Patient Health Questionnaire (PHQ-9) was used to screen for depression. It is made up of 9 Likert-scale questions, which align with the Diagnostic and Statistical Manual of Mental Disorders criteria for major depressive disorder [20]. The total PHQ-9 score was used to categorise the severity of depression into mild (5–9), moderate (10–14), moderately severe (15–19) or severe (20–27). The standard cut-off score of ≥10 was used as being indicative of a major depression disorder. Finally, to measure anxiety, the Generalised Anxiety Disorder Questionnaire (GAD-7) was administered. The total score on this seven-item instrument was used to categorise mild (5–9), moderate (10–14) or severe (≥15) anxiety. A score of ≥10 was used as a cut-off score for possible generalised anxiety disorder [21]. Finally, a QoL of life measure (WHO-QoL-BREF) was used as a proxy for gaining insight into bio-socioeconomic stressors. Comprising of 26 items, this tool captures physical, social, environmental and psychological factors of QoL [22].

Questionnaires were translated and back-translated from English into French. Supported by the field team on site, a priest and a teacher from the local communities translated between French and Tshiluba, the national language in Kasai, and back-translated

to minimise any discrepancies. The Tshiluba survey was then piloted in the communities before finalisation. A minor adaptation to the PHQ-9 was made to reflect the local context: Item 7, 'Trouble concentrating on things such as reading newspaper and watching television' was amended to incorporate more relevant examples - specifically 'listening to radio or church service'.

## Data analysis

Data analysis was completed using SPSS Version 28.0 [23] and R software [24]. Descriptive analysis was used to present sex-disaggregated data on PHQ-9, GAD-7, and QoL, with prevalence rates calculated using recommended cut-off scores where relevant. Reliability of measurement scales was explored using Cronbach's alpha. Mann-Whitney tests were used to explore differences between sexes in continuous scores; Z-tests examined differences between proportions of prevalence of depression and anxiety across waves and across sexes. Multivariable linear regression was applied to examine whether time between surveys, physical and environmental QoL and demographic factors served as predictors for poor mental health. Interaction effects between these predictors and sex were explored. All variables were entered simultaneously. The decision was made a priori to not include the psychological QoL domain as a predictor. Since this item also captures psychopathological symptoms, it was felt that the item content of this domain was tautological and overlapped too much with the mental health outcomes of interest. After initial analysis, the decision was made to also exclude the social QoL domain due to low internal consistency.

## Results

Overall, 200 individuals completed the questionnaire in Wave 1; this was 185 in Wave 2. Table 1 shows the characteristics of study participants. These were similar across both waves; with a marginally higher proportion of men and slightly more people who were not in formal employment (e.g. unemployed or subsistence farmers). The average age was 40 years in Wave 1 and 42 years in Wave 2.

Table 1. Demographic characteristics of study participants.

| Characteristic | Wave 1: 2022 (n = 200) | | Wave 2: 2023 (n = 185) | |
|---|---|---|---|---|
| | n | % | n | % |
| Sex | | | | |
| Male | 107 | 53.5 | 96 | 51.9 |
| Female | 93 | 46.5 | 89 | 48.1 |
| Age (years) | | | | |
| 18–29 | 58 | 29.0 | 40 | 21.6 |
| 30–49 | 88 | 44.0 | 88 | 47.6 |
| ≥50 | 54 | 27.0 | 57 | 30.8 |
| Formal employment | | | | |
| Yes | 91 | 45.5 | 83 | 44.9 |
| No | 108 | 54.0 | 102 | 55.1 |
| Missing | 1 | 0.5 | | |

*Note.* N = 200 in Wave 1 and N = 185 in Wave 2. Participants were on average 40.0 ± 15.0 years old in Wave 1 and 42.0 ± 14.0 years old in Wave 2.

**Table 2. Mental health outcomes across participant characteristics.**

| Characteristic | Wave 1: 2022 (n = 200) | | Wave 2: 2023 (n = 185) | |
|---|---|---|---|---|
| | PHQ-9M (sd) | GAD-7 M (sd) | PHQ-9 M (sd) | GAD-7 M (sd) |
| Overall | 8.4 (5.6) | 7.2 (4.6) | 7.4 (4.6) | 6.6 (3.9) |
| Sex | | | | |
| Male | 7.6 (5.2) | 7.0 (4.4) | 7.0 (4.5) | 6.6 (3.9) |
| Female | 9.5 (5.8) | 7.3 (4.9) | 7.8 (4.7) | 6.6 (3.9) |
| Age (years) | | | | |
| 18–29 | 8.3 (5.8) | 6.8 (4.5) | 5.6 (4.2) | 5.2 (3.2) |
| 30–49 | 8.4 (5.6) | 7.3 (4.3) | 7.6 (4.6) | 4.2 (7.1) |
| ≥50 | 8.7 (5.3) | 7.3 (5.2) | 8.3 (4.5) | 5.9 (3.8) |
| Formal employment | | | | |
| Yes | 8.1 (5.6) | 7.1 (4.6) | 7.3 (4.7) | 6.6 (3.8) |
| No | 8.6 (5.4) | 7.2 (4.5 | 7.5 (4.4) | 6.7 (4.0) |

Table 2 shows the mean PHQ-9 and GAD-7 scores per participant characteristic. The overall mean PHQ-9 score was 8.4 (SD = 5.6, range 0 to 23) in Wave 1 and 7.4 (SD 4.6, range 0 to 20) in Wave 2. As seen in S1 Appendix, using a cut-off score of ≥10, the proportion of study participants screening positive for major depressive disorder was similar across both waves (35.5% in Wave 1 and 33.0% in Wave 2). An additional ~36% of participants met the criteria for mild depression (PHQ-9 ≥5). In Wave 1, 24.0% of participants reported thoughts of self-harm or suicide, this had reduced to 14.6% during Wave 2 (Z = 2.33, $p$ = 0.02). Internal consistency of the PHQ-9 was acceptable (α = 0.71).

Exploring the differences between sexes, the data showed that in Wave 1, a higher proportion of women (43.0%) than men (29.0%) displayed symptoms indicative of major depressive disorder; this difference was no longer significant during Wave 2 (34.8% for women versus 31.3% for men). Notably, 31.2% of women and 17.8% of men had thoughts of being better off dead or harming themselves in the two weeks preceding the Wave 1 survey (Z = −2.22, $p$ = 0.03, [Hyperlink_para10414_29761906112]S1 Appendix[Hyperlink]); this was 19.1% and 10.1% during Wave 2.

The mean GAD-7 score during Wave 1 was 7.2 (SD = 4.6, range 0 to 20) and 30.9% of participants displayed symptoms of moderate or severe anxiety. During Wave 2, the mean score was 6.6 (SD = 3.9, range 0 to 20), with 21.6% of participants displaying anxiety symptoms. There was a significant reduction in anxiety prevalence rates over time (from 30.9% to 21.6%; Z = 2.33, $p$ = 0.04), which appears linked to reduced anxiety amongst women between waves (34.4% to 19.1%; Z = −2.22, $p$ = 0.03). Internal consistency for the GAD-7 was acceptable (α = 0.6). Co-occurrence of depression and anxiety was 22% at Wave 1 and 14% at Wave 2.

Furthermore, the majority of men (64.5%) in Wave 1 rated their QoL as very poor/poor, this reduced to 50.0% at Wave 2. Although still high, women reported less dissatisfaction with QoL than men (37.6% in Wave 1 and 34.8% in Wave 2). The lowest mean-per-item scores ([Hyperlink_para10415_29761906112]S2 Appendix[Hyperlink]) were reported for questions related to finances, dependence on medication, lack of access to healthcare, transport and recreational opportunities. This gives some insight into daily stressors that participants continue to face. Table 3 presents the mean and standard deviation (sd) scores of the four QoL domains. Environmental and physical QoL were rated the lowest across both

**Table 3. Quality of Life scores across waves, per sex.**

| Domain | Wave 1: 2022 | | | Wave 2: 2023 | | |
|---|---|---|---|---|---|---|
| | Overall M (sd) | Men M (sd) | Women M (sd) | Overall M (sd) | Men M (sd) | Women M (sd) |
| Physical ($\alpha$ = 0.63) | 52.2 (16.0) | 49.6 (16.5) | 55.2 (15.0) | 55.9 (16.8) | 53.9 (18.6) | 58.3 (14.4) |
| Environmental ($\alpha$ = 0.52) | 45.8 (17.3) | 43.4 (17.7) | 48.6 (16.4) | 45.6 (15.0) | 43.0 (14.7) | 48.4 (14.9) |
| Social ($\alpha$ = 0.24) | 65.1 (16.4) | 65.5 (18.5) | 64.7 (13.7) | 64.1 (16.7) | 65.9 (18.4) | 61.9 (14.6) |
| Psychological ($\alpha$ = 0.60) | 66.4 (21.0) | 68.6 (18.6) | 63.8 (23.3) | 68.8 (18.6) | 66.4 (16.1) | 71.6 (20.8) |

Waves. Reliability of the overall scale was acceptable ($\alpha$ = 0.78), but unsatisfactory for the social domain (Table 2).

Multivariable linear regression models including demographic data and physical and environmental QoL scores were found to significantly predict depression and anxiety. As seen in Table 4, time was not a significant predictor of depression or anxiety. Instead, depression and anxiety were both predicted by being female, being of older age, and by experiencing lower physical QoL. Significant interaction effects were found between sex and environmental QoL for both outcomes, indicating that environmental QoL significantly predicted depression and anxiety for women, but not for men (Table 4 and Fig 1).

**Table 4. Multivariable linear regression models for depression and anxiety.**

| Predictor | Depression (PHQ-9) (n = 382) | | Anxiety (GAD-7) (n = 382) | |
|---|---|---|---|---|
| | Model 1 | Model 2 | Model 1 | Model 2 |
| | *b* (SE) | *b* (SE) | *b* (SE) | *b* (SE) |
| Time (ref: Wave 1) | −0.73 | −0.71 | −0.30 | −0.28 |
| | (0.48) | (0.47) | (0.41) | (0.41) |
| Age (years) | 0.04** | 0.04*** | 0.03* | 0.03* |
| | (0.02) | (0.02) | (0.01) | (0.01) |
| Employed (ref:no) | 0.42 | 0.38 | 0.32 | 0.29 |
| | (0.50) | (0.49) | (0.43) | (0.42) |
| Sex (ref: male) | 2.46*** | 6.92*** | 1.00** | 3.72*** |
| | (0.51) | (1.47) | (0.44) | (1.27) |
| Physical QoL | −0.09*** | −0.10*** | −0.07*** | −0.08*** |
| | (0.02) | (0.02) | (0.01) | (0.01) |
| Environmental QoL | −0.06*** | −0.01 | −0.05*** | −0.02 |
| | (0.02) | (0.02) | (0.01) | (0.02) |
| Environmental QoL*Sex | | −0.10*** | | −0.06** |
| | | (0.03) | | (0.03) |
| Constant | 12.71*** | 10.95*** | 11.53*** | 10.45*** |
| | (1.26) | (1.36) | (1.09) | (1.18) |
| $R^2$ | 0.20 | 0.22 | 0.16 | 0.17 |
| Adjusted $R^2$ | 0.19 | 0.21 | 0.15 | 0.16 |

***$p < 0.001$, **$p < 0.01$, *$p < 0.05$.

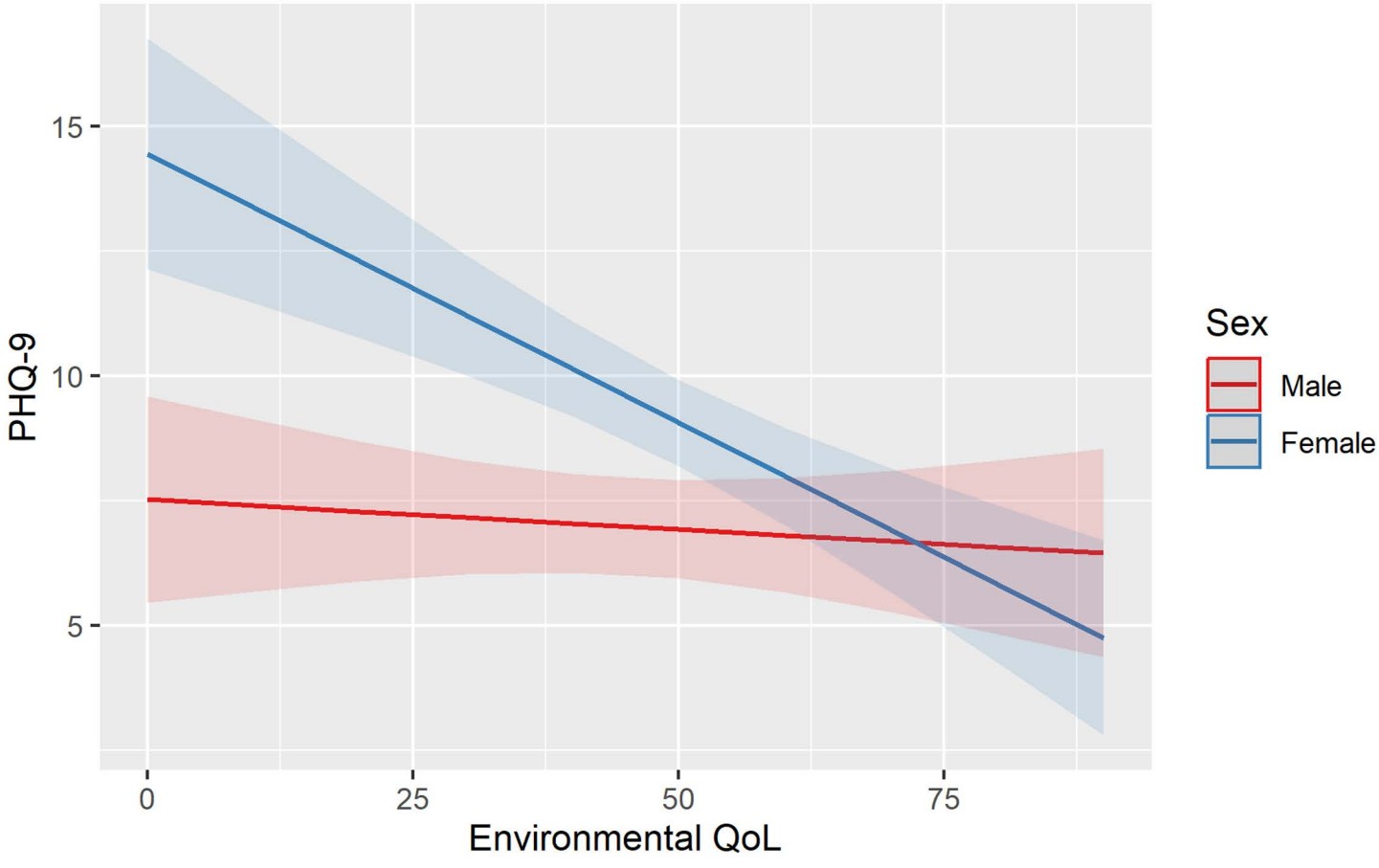

**Fig 1. Interaction effect between sex and environmental Quality of Life for depression.**

## Discussion

This repeated cross-sectional study investigated the prevalence of depression and anxiety in a sample of the general population from the Tshikapa area in the conflict-affected Kasai Province, DRC. Consistent with previous studies from post-conflict settings [6], the findings indicate a large mental health burden. The pooled prevalence across both waves was 34.3% for major depression disorder and 25.8% for generalised anxiety disorder. In addition, 35.8% of participants experienced mild depressive symptoms and 40.2% experienced mild symptoms of anxiety. Detailed comparison of these findings with other studies of mental health in conflict-affected populations is difficult due to variation in type of target population, assessment methods, and time elapsed since conflict end. We identified two other studies from the DRC with conflict-affected persons in the east of the country. One study found slightly lower rates of depression (27.8%) and similar rates of anxiety (25.4%) [25], whilst the other reported a higher prevalence of major depressive disorder of 40.5%. However, the latter study used a one-year recall period and applied a different scoring method to PHQ-9 questions [12]. Global WHO prevalence estimates in post-conflict settings were 10.8% for mild to severe depression and 21.7% for mild to severe anxiety [6]. Particularly for depression, the rates identified in the current study were higher than these estimates. Whilst we recognise that this may be due to limitations with prevalence overestimates when utilising self-reported tools

[6], there is clearly a significant need for the development of currently lacking mental health services in the region.

This study used a QoL measure as a proxy to gain further insight into the impact of biosocial factors on mental health. It showed that generally, even though the conflict has ended, participants continue to be exposed to a multitude of daily stressors related to poverty, insecurity, and poor physical health. These factors are likely to be both a cause and consequence of the protracted conflict in the region. Multivariable regression analysis found that at one year between measurements, time was not a significant predictor of mental health outcomes. Instead, the data indicated that prevalence rates may remain high if post-conflict populations do not have access to mental health services and continue to be exposed to daily stressors (which might have already been present before the conflict). Consistent with other studies [6], depression and anxiety were both predicted by female sex, increased age, and lower physical QoL. For women, but not for men, environmental QoL – which relates to aspects of personal safety, financial security, and access to health services – was found to predict poor mental health outcomes. Significant improvements in feelings of safety amongst women between both Waves might account for the reduction in female anxiety levels that was highlighted in univariable analysis.

This study took a broad approach to examining the relationship between conflict and mental health in the current social environment. It did not measure experiences (e.g., exposure to violent events and internal displacement) and their associations with mental health outcomes (e.g., PTSD). This limits how much this paper can contribute to the debate around the magnitude of direct effects of war exposure in explaining psychological distress, and the mediating role of daily stressors in the relationship between these [4]. However, there are ongoing debates within the humanitarian sector as to the added public health value of significant focus being placed on PTSD. It is argued that the accompanying emphasis on trauma-focused interventions might be at the expense of a broader psychosocial approach to promoting mental wellbeing and human development. Further (qualitative) research would be needed to explore the causal direction of the relationship between stressors and mental health outcomes identified and the moderating properties of conflict.

Still, this paper provides important insights that could guide data-informed decision making for the funding and commissioning of mental health services within DRC. The findings support the growing consensus that such services should go beyond biomedical interventions and include approaches that consider social determinants of (mental) health and provide multi-layered support [26,27]. Ensuring the provision of basic services and security as well as more focused mental health support is essential, including: 1) Interventions that improve social and material conditions caused or worsened by the conflict such as breakdowns in infrastructure or reduction of livelihood opportunities, addressed through multi-sectoral collaboration and population-based initiatives; 2) broader family and community-based psychosocial programmes that focus on social support and strengthening community relationships (e.g., community healing dialogues [28]) following conflict; 3)programmes focusing on mental health promotion and self-care for emotional health and focused but non-specialised support for those with lower-level mental health conditions, or those who may have experienced traumatic events that require further intervention, for example gender based violence; and 4) the provision of specialised services for those with more complex mental health conditions. The need for the latter two levels of interventions is particularly evident through the sustained high levels of distress and high numbers of individuals indicating they have suicidal ideation or a tendency to self-harm; the data provided in Appendix 1 and 2 could support policy makers with initial estimates of the proportion of people requiring individual support and main target areas for community-level interventions. Whilst we did not capture experiences of gender inequities, our sex-disaggregated analysis, paired with the knowledge of

a persistent gender crisis in the province [16], indicate that these interventions should focus on taking steps toward gender-transformative change and include approaches to tackle discriminatory gendered norms and barriers to service access. There have been few intervention studies in DRC. We identified none in the Kasai region, but four group therapy and family support interventions, and two socio-economic interventions, mostly in eastern DRC, have so far shown promising results [29].

As a further limitation, our study did not include children and adolescents. Many child soldiers (boys and girls) were recruited during the conflict and further research is needed to understand impact of the conflict on mental health of young people in the region. Whilst this study was not a longitudinal study that collected data from the same participants over time, the repeated cross-sectional design of this study has provided interesting insight into the effect of time on mental health outcomes; we believe that this is a significant strength of the study. Furthermore, our sex-disaggregated data will be useful in gender-responsive policy and programme action. Finally, an important methodological detail is that the survey had to be administered verbally due to low literacy rates. The impact of this on reported prevalence rates requires further investigation.

## Conclusion

To our knowledge, this is the first study to assess mental health in a conflict-affected population of the Kasai Province in DRC. Measured twice, with a one-year interval, this study consistently found a high prevalence of depression, anxiety and suicidal thoughts. Women were found to be at increased vulnerability to developing mental health conditions Participants reported a low QoL, providing insight into their daily stressors. The data indicated that the passing of time did not significantly predict mental health. Instead, daily stressors that are either caused or exacerbated by conflict (e.g., poverty, lack of access to healthcare) continue to affect mental health after a conflict has ended. Approaches that go beyond psychological and biomedical mental health treatments to include multi-sectoral interventions that address broader social determinants of (mental) health are recommended.

## Supporting information

**S1 Appendix. Mental health outcomes across waves and sex.**
(DOCX)

**S2 Appendix. QoL mean-per-item scores, per sex.**
(DOCX)

**S1 Data. Dataset.**
(SAV)

**S2 Data. Analysis codes.**
(R)

**S1 Checklist. Inclusivity in global research questionnaire.**
(DOCX)

## Acknowledgments

We would like to thank Benoit Tshishiku, Clement Mukendi, Antoinette Mbokashanga, Naomie Sengu, Atishimen Bakamba, Florance Ntumba, Dimuenayi Marcelo, Garcia Chita, Clement Kande and Didier Mubudi for their contributions to the data collection process. In addition, we would like to thank Dr Lucas Sempe for his support with data analysis.

## Author contributions

**Conceptualization:** Maaike L. Seekles, Pierre O. L. Omumbu, Joy J. Kim, Raphael M. Mulamba, Motto Nganda, Laura Dean.

**Data curation:** Maaike L. Seekles, Jacob K Kadima, Pierre O. L. Omumbu, Junior K. Kukola, Joy J. Kim, Christian B Bulambo, Raphael M. Mulamba, Motto Nganda, Laura Dean.

**Formal analysis:** Maaike L. Seekles.

**Funding acquisition:** Christian B Bulambo, Raphael M Mulamba, Laura Dean.

**Investigation:** Maaike L. Seekles, Jacob K Kadima, Pierre O. L. Omumbu, Junior K. Kukola, Joy J. Kim, Christian B Bulambo, Raphael M Mulamba, Motto Nganda, Laura Dean.

**Methodology:** Maaike L. Seekles, Jacob K Kadima, Pierre O. L. Omumbu, Junior K. Kukola, Joy J. Kim, Raphael M Mulamba, Motto Nganda, Laura Dean.

**Project administration:** Joy J. Kim, Christian B Bulambo, Raphael M Mulamba, Laura Dean.

**Resources:** Joy J. Kim, Christian B Bulambo.

**Software:** Maaike L. Seekles.

**Supervision:** Laura Dean.

**Validation:** Maaike L. Seekles.

**Visualization:** Maaike L. Seekles.

**Writing – original draft:** Maaike L. Seekles, Laura Dean.

**Writing – review & editing:** Maaike L. Seekles, Jacob K Kadima, Pierre O. L. Omumbu, Junior K. Kukola, Joy J. Kim, Christian B Bulambo, Raphael M Mulamba, Motto Nganda, Laura Dean.

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
