## [Decision Letter · Decision Letter 0]

12 Dec 2023

PGPH-D-23-01634

Prevalence of mental health conditions in post-conflict Kasai Province, Democratic Republic of the Congo: a repeated, cross-sectional study.

Dear Dr. Seekles,

Thank you for submitting your manuscript to PLOS Global Public Health. After careful consideration, we feel that it has merit but does not fully meet PLOS Global Public Health’s publication criteria as it currently stands. Therefore, we invite you to submit a revised version of the manuscript that addresses the points raised during the review process.

We look forward to receiving your revised manuscript.

Kind regards,

Vanessa Carels

Staff Editor

Journal Requirements:

3. Please provide separate figure files in .tif or .eps format.

4. We have noticed that you have uploaded Supporting Information files, but you have not included a list of legends. Please add a full list of legends for your Supporting Information files after the references list.  

Additional Editor Comments (if provided):

Reviewers' comments:

Reviewer's Responses to Questions

**Comments to the Author**

1. Does this manuscript meet PLOS Global Public Health’s publication criteria ? Is the manuscript technically sound, and do the data support the conclusions? The manuscript must describe methodologically and ethically rigorous research with conclusions that are appropriately drawn based on the data presented.

Reviewer #1: Partly

Reviewer #2: Yes

2. Has the statistical analysis been performed appropriately and rigorously?

Reviewer #1: Yes

Reviewer #2: Yes

3. Have the authors made all data underlying the findings in their manuscript fully available (please refer to the Data Availability Statement at the start of the manuscript PDF file)?

Reviewer #1: Yes

Reviewer #2: Yes

4. Is the manuscript presented in an intelligible fashion and written in standard English?

Reviewer #1: Yes

Reviewer #2: Yes

5. Review Comments to the Author

Reviewer #1: Overall

The study addresses the mental health symptoms in a conflict-torn area of Eastern DRC. The conflict lasted for three years (2016-2019) and a repeated cross-sectional measurement was carried out in 2022 and 2023. The purpose seems to be two-fold: (i) to inform future interventions or programming to address the mental health problems, and (ii) to study the impact of bio-socioeconomic stressors, using the environmental domain of the WHO-QoL Bref instrument.

Overall, the study and its two purposes are justified. Another bigger population study in the same region is referred to (ref 18), but this study collected data in 2017, i.e. during the conflict, while the data in this study was collected 3 and 4 years after the end of the conflict.

The methods applied for the basic assessment of prevalence of mental health symptoms including sampling and analyses seem appropriate. However, it is not fully clear whether the data and the methods match the two purposes raised.

Major issues

No data on conflict related traumatic events are presented. This leaves the reader with two questions: on (i) how these proxy indicators of bio-socioeconomic stressors included should be interpreted: Are they remaining conflict related factors or are they common indicators of poverty that were also present before the conflict? If so, how do they – if at all - interact with conflict related mental health outcomes? And (ii) could the assessment of these stressors possibly be associated with the indicators of mental health (clearly stated by the authors for the psychological domain but not for the other three) so that concerns about safety, economy, access to health and transport are rather manifestations of anxiety and depression than the cause of them? It requires a more rigorous argumentation, addressing these two questions, to present the association between mental health and the WHO-QoL Bref environment domain (applying for women only) as a relevant finding.

Secondly, on the information of interventions: It is somewhat diffuse how the findings may contribute to future programming or interventions (multi-sectoral interventions addressing broader social determinants). The paper would gain from more clarity in terms of what contributions the study makes.

Minor issues

There are not many considerations about the ethical implications of undertaking surveys in conflict affected populations. It seems pertinent to describe which information of participation risks was communicated to potential participants, if any participants seemed to have been re-traumatized, if interviewers were capable of handling such situations, and if any referrals to further support or treatment were made.

The paper states that 1 million people were internally displaced (IDP), but not whether they had returned by 2022 and 2023 and not either what proportion of the study population were IDP. It could be assumed that the environmental domain would be related to IDP in some way.

Reviewer #2: 1.Discussion could include more information about the qualitative aspects of life for the population in general and women in particular.

2.The area for inclusion is the aspects SELF CARE For EMOTIONAL HEALTH. This is an emerging area in disaster mental health.

6. PLOS authors have the option to publish the peer review history of their article (what does this mean? ). If published, this will include your full peer review and any attached files.

**Do you want your identity to be public for this peer review?** For information about this choice, including consent withdrawal, please see our Privacy Policy .

Reviewer #1: No

Reviewer #2: **Yes: ** Srinivasa Murthy R

---

## [Decision Letter · Decision Letter 1]

4 Jul 2024

PGPH-D-23-01634R1

Prevalence of mental health conditions in post-conflict Kasai Province, Democratic Republic of the Congo: a repeated, cross-sectional study.

Dear Dr. Seekles,

Thank you for submitting your manuscript to PLOS Global Public Health. After careful consideration, we feel that it has merit but does not fully meet PLOS Global Public Health’s publication criteria as it currently stands. Therefore, we invite you to submit a revised version of the manuscript that addresses the points raised during the review process.

We look forward to receiving your revised manuscript.

Kind regards,

Feten Fekih-Romdhane

Academic Editor

Journal Requirements:

Additional Editor Comments (if provided):

Reviewers' comments:

Reviewer's Responses to Questions

**Comments to the Author**

1. If the authors have adequately addressed your comments raised in a previous round of review and you feel that this manuscript is now acceptable for publication, you may indicate that here to bypass the “Comments to the Author” section, enter your conflict of interest statement in the “Confidential to Editor” section, and submit your "Accept" recommendation.

Reviewer #1: All comments have been addressed

Reviewer #3: All comments have been addressed

Reviewer #4: All comments have been addressed

Reviewer #5: (No Response)

2. Does this manuscript meet PLOS Global Public Health’s publication criteria ? Is the manuscript technically sound, and do the data support the conclusions? The manuscript must describe methodologically and ethically rigorous research with conclusions that are appropriately drawn based on the data presented.

Reviewer #1: Yes

Reviewer #3: Yes

Reviewer #4: Yes

Reviewer #5: Yes

3. Has the statistical analysis been performed appropriately and rigorously?

Reviewer #1: Yes

Reviewer #3: Yes

Reviewer #4: Yes

Reviewer #5: Yes

4. Have the authors made all data underlying the findings in their manuscript fully available (please refer to the Data Availability Statement at the start of the manuscript PDF file)?

Reviewer #1: Yes

Reviewer #3: Yes

Reviewer #4: Yes

Reviewer #5: Yes

5. Is the manuscript presented in an intelligible fashion and written in standard English?

Reviewer #1: Yes

Reviewer #3: Yes

Reviewer #4: Yes

Reviewer #5: Yes

6. Review Comments to the Author

Reviewer #1: None

Reviewer #3: General comments and some key concerns:

1. It is an interesting study that is giving an insight on the “Prevalence of mental health conditions in post-conflict Kasai Province, Democratic Republic of the Congo: a repeated, cross-sectional study.”

•Suggest “conditions” be replaced with “disorders”. Apply this in the entire document.

•Replace “ sample” with “study participants” all through the document

•Avoid long sentences as the meaning get lost – Use short sentences

•Percentages be reported to 1 decimal place

•Tables should converted to Scientific tables

•Use term “Study participants” other than “sample” all through the document

2. Abstract

•Much as it is a block abstract, the following sub-sections should be clear in it (Background, aim, methods, results and conclusion)

•Background information in Mental health disorders in Kasai Province, Democratic Republic of the Congo is missing

•Aim of the study is missing

•Line 34: Percentages be reported to 1 decimal place i.e. 34.0%

•Conclusion is missing

•Use third person in sentence construction – see line 85

3. Introduction

•Line 46: Write it as a complete sentence and put a citation on it

•Provide, a highlight on the various mental illnesses in the region

•Elaborate more on how conflict can lead to mental illnesses with examples

•Besides, conflict; are there other risk factors that that would exacerbate mental illness in the region that need to be highlighted

4. Methods section

•Line 89: should be Methodology instead of Methods

•Line 90: Design and sample should be separated and should be written as “ Study design”

•In this sub-section, ensure these sections are clear in the Methodology

oStudy design

oStudy setting

oStudy population

oSample size

oSelection criteria -both inclusion and exclusion-criteria

oStudy participant recruitment or enrollment

oTool used

oStudy procedures

oData management and quality control

oStatistical data analysis

oEthical consideration -issue of consent forms and a statement on Helsinki declaration should be included

5. Results

•Line 149: The average age was around 40 years in both samples; Was it net 40 years or 40.---± -- years?

•Line 150: Should be written as “Demographic characteristics of the study participants and mental health outcome scores”

•Though an independent table on detailed “Demographic characteristics of the study participants” should be provided

•Table 1: The figures should quoted as % (n) other than n (%) as the percentage is the main measure and not frequencies

•The frequencies such as Formal employment do not add up to the total 200 study participants and please check the figures all through the tables or account for the missing participants

•Table 1 should be converted to Scientific table

•Report overall mean as: Mean±--- other than Mean (SD) and this should be done all through the tables

•Line 152: What is the study characteristic?

•In table 2: The values in the rows with brackets (); that do they mean?

•Where are the p-values and confidence intervals?

6. Discussion

•Should be sub-sectioned according the results sub-sections or domains such as:

oDemographic characteristics of the study participants

oDemographic characteristics of the study participants and mental health outcome scores

oQuality of Life scores

oMultivariable linear regression models for depression and anxiety

•Line 204: Provide citation for the reported other study

•Percentages should be written to one (1) decimal place and all through the document

•What were the limitations of the study?

Conclusion

•Let the conclusion be drawn on all the 3 domains of the study

Reviewer #4: Thank you for correcting and reviewing the recommendations previously made by other reviewers. This article is good as it is, but it could be better.

Format comments:

1. Add reference in line 25

2. Correct Organisation to Organizartion in line 59

3. Correct "data was" instead of "data were" in line 100

4. Line 147, it isn't clear if the correct term is "both surveys" or "both waves", please review

5. In line 229 add "on" before PTSD

Content comments:

It isn't clear, and I think it's an important variable, if the group mental health session was held before or after the survey applications. If it was before, it is important to note as this might have helped participants to better answer the surveys, and this is important for discussion. If it happened after the surveys, this could help explain why the prevalence rates reduced after time?

If there were literacy issues with the participants, were the surveys administered with someone else and they had to give out the answer? Please clarify and state this clearly. This is important as well because if this was done with the help of someone else, the answers could have been influenced by the need of the participants to not appear so negative and give out negative responses (specially after 1 year).

Please give further explanation as why you think the prevalence rates reduced on wave 2, this is such an interesting finding!!!

Good luck!

Reviewer #5: Dear Authors, this paper is well written and pertains a vital part of mental health in conflict area DRC which would affect the livelihood of the population. There are some minor revisions that need to be corrected before publication.

Background; you should integrate other studies done within the East African region like Northern Uganda and South Sudan or even Burundi with the recent conflicts. There have also been studies done to assess the mental health of refugees from the DRC in Uganda. that would help to add weight and relevance to the paper. There have also been studies that suggest the mental health interventions that have been made within the country, Could you also highlight those and the interventions in place and if they could affect your paper.

Discussion; the discussion states that time was not a significant aspect in the PHQ and anxiety scale scores but were there some interventions by either local government or organizations during the time period in the study areas that need to be noted or the status of the community remained the same within the one year time period that could have affected your results.

7. PLOS authors have the option to publish the peer review history of their article (what does this mean? ). If published, this will include your full peer review and any attached files.

**Do you want your identity to be public for this peer review?** For information about this choice, including consent withdrawal, please see our Privacy Policy .

Reviewer #1: No

Reviewer #3: No

Reviewer #4: No

Reviewer #5: No

---

## [Decision Letter · Decision Letter 2]

14 Oct 2024

PGPH-D-23-01634R2

Prevalence of mental health conditions in post-conflict Kasai Province, Democratic Republic of the Congo: a repeated, cross-sectional study.

Dear Dr. Seekles,

Thank you for submitting your manuscript to PLOS Global Public Health. After careful consideration, we feel that it has merit but does not fully meet PLOS Global Public Health’s publication criteria as it currently stands. Therefore, we invite you to submit a revised version of the manuscript that addresses the points raised during the review process.

We look forward to receiving your revised manuscript.

Kind regards,

Feten Fekih-Romdhane

Academic Editor

Journal Requirements:

Additional Editor Comments (if provided):

Reviewers' comments:

Reviewer's Responses to Questions

**Comments to the Author**

1. If the authors have adequately addressed your comments raised in a previous round of review and you feel that this manuscript is now acceptable for publication, you may indicate that here to bypass the “Comments to the Author” section, enter your conflict of interest statement in the “Confidential to Editor” section, and submit your "Accept" recommendation.

Reviewer #1: All comments have been addressed

Reviewer #3: All comments have been addressed

Reviewer #5: All comments have been addressed

Reviewer #6: All comments have been addressed

2. Does this manuscript meet PLOS Global Public Health’s publication criteria ? Is the manuscript technically sound, and do the data support the conclusions? The manuscript must describe methodologically and ethically rigorous research with conclusions that are appropriately drawn based on the data presented.

Reviewer #1: Yes

Reviewer #3: Yes

Reviewer #5: Yes

Reviewer #6: Yes

3. Has the statistical analysis been performed appropriately and rigorously?

Reviewer #1: N/A

Reviewer #3: Yes

Reviewer #5: Yes

Reviewer #6: Yes

4. Have the authors made all data underlying the findings in their manuscript fully available (please refer to the Data Availability Statement at the start of the manuscript PDF file)?

Reviewer #1: Yes

Reviewer #3: Yes

Reviewer #5: Yes

Reviewer #6: Yes

5. Is the manuscript presented in an intelligible fashion and written in standard English?

Reviewer #1: (No Response)

Reviewer #3: Yes

Reviewer #5: Yes

Reviewer #6: Yes

6. Review Comments to the Author

Reviewer #1: (No Response)

Reviewer #3: Tables should converted to Scientific tables (Editor will decide)

Line 133: Statement “Questionnaires were translated and back-translated from English into French”. Why was it translated in English?

Line 160: Table 1 Demographic characteristics of study participants should be “Table 1: Demographic characteristics of study participants”

Table 1: Age should have a unit i.e. Age (years)

Line 161: average 40.0 years old (SD = 15.0) should be written as “40.0±15.0 years”

Table 2: Age should have a unit i.e. Age (years)

Table 4: Age should have a unit i.e. Age (years)

Line 205: Discussion – to be consistent, the Percentages be reported to 1 decimal place i.e. 34.0% and not 34%

Reviewer #5: None.

Congratulations upon a well written manuscript.

Reviewer #6: This study is a repeated cross-sectional study that investigated the prevalence of depression and anxiety in a post-conflict population in Kasai Province, DRC. This is important because the findings could guide data-informed decision making for the funding and commissioning of mental health services within DRC.

I also agree that the authors' responses to the reviewers' comments were thorough and well-considered. Using "condition" instead of "disease" is appropriate given that formal diagnoses were not made, and the minor points you raise are valid and should be addressed as follows:

a) Line 120: The sentence "...School of Tropical Medicine (reference number: (21-053)." is missing a closing parenthesis. Please add a closing parenthesis after "21-053". Alternatively, you could rewrite the phrase as "(reference number: 21-053)."

b) The journal name in reference 14 should be "JAMA", not "Jama".

7. PLOS authors have the option to publish the peer review history of their article (what does this mean? ). If published, this will include your full peer review and any attached files.

**Do you want your identity to be public for this peer review?** For information about this choice, including consent withdrawal, please see our Privacy Policy .

Reviewer #1: No

Reviewer #3: No

Reviewer #5: No

Reviewer #6: No

---

## [Editor Report · Decision Letter 3]

26 Nov 2024

Prevalence of mental health conditions in post-conflict Kasai Province, Democratic Republic of the Congo: a repeated, cross-sectional study.

PGPH-D-23-01634R3

Dear Dr Seekles,

We are pleased to inform you that your manuscript 'Prevalence of mental health conditions in post-conflict Kasai Province, Democratic Republic of the Congo: a repeated, cross-sectional study.' has been provisionally accepted for publication in PLOS Global Public Health.

Best regards,

Feten Fekih-Romdhane

Academic Editor